# Developing a Wine Experience Scale: A New Strategy to Measure Holistic Behaviour of Wine Tourists

**Vasco Santos** [1,2,3,*], **Paulo Ramos** [4,5], **Nuno Almeida** [2] **and Enrique Santos-Pavón** [6]

1   University of Seville, Calle San Fernando, 4, 41004 Sevilla, Spain
2   CiTUR, ESTM, Polytechnic of Leiria, Rua do Conhecimento, 2520-641 Peniche, Portugal; nunoalmeida@ipleiria.pt
3   ISLA Santarém, Largo Cândido Reis, 2000-24 Santarém, Portugal
4   CBQF UCP–ESB, Rua de Diogo Botelho, 1327, 4169-005 Porto, Portugal; pramos@ufp.edu.pt
5   Fernando Pessoa University, Praça 9 Abril, 349, 4249-004 Porto, Portugal
6   Department of Physical Geography and Regional Geographical Analysis of the University of Seville, 41004 Sevilla, Spain; esantos@us.es
*   Correspondence: vasco-rs@hotmail.com

**Abstract:** This study develops a scale to measure wine tourism experiences and was tested in Portugal, in two of the main wine tourism centres: Porto and Madeira. The wine experience scale combines experience traits with the traditional approach to scales related to wine tourism. The development of the scale follows the most recognised validated procedures. Data were collected from a total of 647 international wine tourists in the wine cellars of the two main fortified wine tourism regions visiting areas: Porto and Madeira. Structural equation modelling (SEM-AMOS) was used as the main analysis and validation tool. The resulting 18-item wine experience scale comprises four major dimensions: (1) Wine storytelling, (2) wine tasting excitement, (3) wine involvement, and (4) winescape. All these showed reliable and validated indicators. This new scale presents a valid new tool to better measure and evaluate experiences in a wine tourism setting. This study offers a broad range of use for academics, managers, planners, and practitioners. It shows how a new measurement tool focused on the wine tourism experience in terms of several outcomes and applications, addressing important practical managerial implications, can have an impact on academic research. Most previous tourism scales still fail to measure the specifics of wine settings. This is the first scale that comprises the dimensions of experience with wine senses, applied in a relevant wine destination where research is still limited. The results are relevant in boosting the increasingly recognized awareness of Portugal as wine tourism, as well as bringing experience scales to the body of knowledge.

**Keywords:** scale validation; SEM; wine storytelling; wine tasting excitement; wine involvement; winescape

## 1. Introduction

Portugal is recognised as a wine tourism destination and has growth potential. In 2019, the tourism revenue contributed 8.7% of the national GDP, with an increase of 8.1% in tourism revenue growth [1,2]. Wine tourism directly contributes to the wine regions' economic development [3]. The 4th UNWTO Global Conference on Wine Tourism (2019), themed 'Co-creating Innovative Experiences', sought to further explore issues related to wine tourism experience for international comparability between destinations. Although Porto and Madeira wines are internationally renowned, there is still limited empirical research on its experience-based wine tourism. It is noteworthy that the Porto wine vines in Douro became the first wine-growing area in the world to be legally regulated in 1756, although the

name Porto was already in use from at least 1619. The history of Madeira wine is at least 200 years old, although vines had been planted since the fifteenth century by order of Henry the Navigator. This makes these two wine regions the most historically significant regions for fortified wines not only in Portugal, but also globally. They are both fortified wines, which means that they are wines to which brandy was added during its winemaking process, normally for conservation and strengthening purposes. They are part of a broader family that includes Sherry, Marsala, Vermouth, and the also Portuguese Moscatel de Setubal. Portugal is the 11th biggest wine producer but the 9th wine exporter in 2018, with an increase of 5% in volume (3 million hl) and an increase of 11% in value (0.8 billion €) [4]. The Porto wine, after some years of declining sales, had, in 2019, an overall increase in sales of 2.5% in volume, although in value it decreased −1.5%. In the domestic market, it had an increase of 3.6% in value despite a decrease of −0,5% in volume. However, in 2020, in the midst of the Covid-19 pandemic crisis, between January and June, it declined −12.4% in volume and −15.4% in value, compared with the same period last year [5]. The Madeira wine also had a decrease of −15% in volume and −19% in value over the same period. This wine had its best year ever in 2018 with over 19.2 million € in value, but with a decrease of −2.9% in value and −6% in volume already in 2019 [6].

In the context of wine tourism, wine-related experiences are a central concept in which the determinants of the success of a wine region emerges through the selection of the customer hedonic concept as an indicator [7]. Wine products offer a wide range of different experiences including wine-related travel, known as an indicator of the wine tourism experience. Wine tourists may travel in search of specific wine tourism experiences, such as visits to the cellars, wineries, vineyards, wine tasting rooms, and/or wine hotels in order to experience an amalgam of different wine-related activities. Therefore, the wine tourism experience comprises the interplay of many factors such as wine tastings, staff, cellar door visits and sales, entertainment, education, and aesthetics [8].

As Hall et al. [9]. argued, wine is seen as an imperative component of the attractiveness of a wine destination. These activities include a set of opportunities in different life domains based on lifestyle and personal experience [10] (p. 152) the opportunity to purchase wine and to learn more about wine [11]. and other wine related matters [12,13]. opportunities for social interaction [11], and communing with others and the opportunity to relax [12,13]. "A favourable winery experience eventuates when a wine tourist engages in a positive interaction with these wine attributes [14] (p. 1). Wine tourism consists of a wine-related activity that integrates wine culture and heritage, providing a dynamic and versatile experience through the visit context [15]. A visit to a wine cellar includes an aesthetic appreciation of the natural atmosphere, the wine cellar, the cultural and historical context of the wine region, production methods, the search for education and diversity, a sense of belonging to the cellar, and the search for authenticity [16]. To Brás, Costa, and Buhalis, "wine regions can establish themselves as destinations through the full integration of different products combining main attractions: from wine and food to accommodation, events and entertainment activities and many other regional services" [17] (p. 1625).

Globally, the dominant literature still has some research gaps in the field of the wine-related touristic experience as the central product and activity of wine tourism. Despite frequent references to the wine tourism experience, when it is associated with the wine experience construct, it remains fragmented. Within the literature, there is still no universally accepted scale that encapsulates all the dimensions of the wine experience. This makes it difficult to examine what attributes and variables should form it. The proposal of a wine experience construct containing the following dimensions: (1) Wine storytelling, (2) wine tasting excitement, (3) wine involvement, and (4) winescape, appears to demonstrate how wine-related experiences occur simultaneously in the context of a visit. This is the first study that demonstrates the combined used of wine experience dimensions to construct a useful measurement tool, applied to wine experience in wine tourism destinations and wine regions. This measurement approach extends the scope of the existing literature, as there is no scale that measures the wine experience of wine tourists. For instance, there is a lack of consensus about how the wine experience occurs and is perceived in the context of wine tourism activities. Hence, there is an

emerging need to develop and validate a new scale addressing the wine experience [18]. The original contribution of the paper is to showcase the dimensions that form the wine experience construct, providing its associated originality and the value added. A literature review related to the wine experience constructs follows, and an overview of the construction, development, and validation of the wine experience scale is described and discussed. Finally, the results, implications, and future research directions are discussed.

## 2. Literature Review: Wine Tourism Experience and the Domain of Constructs

### 2.1. Wine Tourism Experience

For tourists, food-and-wine activities are a component of their tourism experience while travelling [19]. and a wine tourism activity involves the participation of a group of individuals seeking experiences related to wines and wineries within wine tourism destinations [20]. Charters and Ali-Knight suggest that "the wine tourism experience can be provided in many ways, the most notable being events and festivals, cultural heritage, gastronomy, hospitality, education, tastings and wine houses, wine sales at cellars/wine houses and winery tours" [21] (p. 312). To Pikkemaat et al. [22], the wine tourism sector has the potential to create experiences for the tourist, especially those looking for historical and cultural values in iconic places, who appreciate genuine experiences, and who are interested in wine, vineyard crops, wine houses, and what the landscape offers [23]. The creation of a tourism experience can be combined through food and culture, for instance in music festivals within wineries [24]. A holistic wine experience occurs mainly in the context of a winery visitation, where the tourist experience has a positive effect on their future behaviour intentions [25,26]. The importance of wine tourism and a hedonic experience is supported by Bruwer and Rueger-Muck [7], who advanced that five wine tourist drivers: (1) Taste wine; (2) buy wine; (3) experience the atmosphere; (4) learn more about wine; and (5) find a unique wine, work to achieve a memorable wine tourism experience at a winery cellar door. Thanh and Kirova [3]. also concluded that experiences are globally positive, and that education and entertainment are relevant when comparing aesthetics and escapism. It is also highlighted that a holistic perspective focuses on the visitors' experience in relation to wine tourism activities and wine regions. Wine tourism is recognised as a holistic experience comprising of a set of wine region features [27]. provided mainly by tasting, cellar door, cellar door sales, and winery tours, among others [21]. Creating memorable experiences, especially in a new wine region, is the culmination of a several unique experiences [28].

The inclusion of wine experience dimensions (wine excitement, wine sensory appeal, winescape, wine storytelling, and wine involvement) is justified as other measurements of wine experience are not just centred on a holistically transversal and also aggregating approach, but encapsulate various stages during a wine tourism visit, allowing a clearer vision of the wine experience. The experiential perspective of wine tourism [23,29] can be enhanced through hedonistic components that characterise wine [9]. In addition, Gómez, Pratt, and Molina [30] revealed that there has been an increase in theory building which highlights the complexity underlying the wine tourism experience and, by extension, to the experiential wine tourist. As such, for the final achievement of the following described dimensions, some of the dimensions derived from the dominant literature were included, others disregarded, and others added, considering the underlying holistic component. The dimensions of existing scales are not directed towards the nature of the wine and wine tourism experience. Consequently, a new scale is necessary, as no current scale objectively measures the wine experience. Within this context, this new scale establishes the most effective symbiosis of the dimensions that mirror the various stages of a wine tourism visit. Accordingly, the scale intercepts the main inherent dimensions for a better acquaintance of the holistic and hedonic perspectives of wine and wine tourism experience, which will yield a richness to both conceptual and theory-building research in this field and prove to be useful in wine tourism.

### 2.2. Wine Excitement

Eating experiences, including the drinking of wine, may convey emotions such as excitement and attract tourists who desire excitement and novelty [31]. Fields [32] and Kim, Eves, and Scarles [33] have indicated that eating local food for the first time is an exciting experience within a destination. Fields [32] demonstrated that physical motivators may also be associated with the opportunity to taste new and exotic foods, and thus local wines may also be part of this experience. Additionally, the exciting experience, while considered as one of the key physical motivators, can be regarded as an event that has excitement as the crucial feature in a leisure activity setting [34]. The place experience is determined by the relationships that exist between tourists, in terms of place excitement and engagement [35]. Kim and Eves [36] also assumed excitement as a motivation to taste local food. Within this context, wine tourists are wine consumers looking for pleasurable winery attractiveness [37], which forms part of the memorable wine tourism experience described by Bruwer and Rueger-Muck.

### 2.3. Wine Sensory Appeal

Customer experience in tourism also comprises sensory components [38]. The literature highlights multi-sensory stimuli and impressions to understand tourist experiences, and that tourists may be attracted towards a destination by visual elements [39–41]. Brochado, Stoleriu, and Lupu [42] suggest that wine tourists accord great value to the multisensory aspect of wine, and they identified twelve themes of sensory experience within Douro wineries: (1) Wine, (2) view, (3) staff, (4) room, (5) hotel, (6) food, (7) restaurant, (8) pool, (9) service, (10) Douro, (11) delicious (food and wine), and (12) comfort. Wine tourism indulges the senses in the wine product itself primarily, involved through the very nature of wine tourism, and influences consumer attitudes and purchases within wineries [43]. Bouzdine-Chameeva and Durrieu [44] suggest sensory stimulation originates in the wine tasting and the winery design. Ali-Knight and Carlsen [45] state that consumer engagement is achieved by novelty and sensory activities in winery settings and was confirmed by Santos el al. [18] where sensory impressions impacted on the winery visit experience.

### 2.4. Winescape

The winescape is described as the synergic interaction of "vineyards, wineries and other physical structures, wines, natural landscape and setting, people and heritage, towns and their architecture and artefacts within them" [46] (p. 277). Alebaki and Lakovidou [47] (p. 123) describe winescape as "the whole region and its attributes". Thomas, Quintal, and Phau [14] also conceptualised seven key attributes of the winescape: (1) The winescape cluster, (2) the atmosphere, (3) the wine product, (4) complementary products, (5) the signage, (6) the layout, and (7) service staff attributes. Dimensions of the winescape include: (1) Nature-related; (2) wineries and vineyards; (3) wine and other products; (4) ambient factors; (5) signage and layout; (6) service staff and locals; (7) heritage-related towns; and (8) fun-based activities [48]. The winescape is also the primary driver of motivations for the wine tourists' hedonic experience [23] where much importance is placed on the winescape during the visit [49]. Bruwer and Gross [50] advocate that a winescape framework for wine tourism is conceptualised by five major dimensions: Infrastructure, natural setting, atmosphere, layout, and people. The winescape attributes shown above are considered in a multi-layered macro-context of a wine region.

### 2.5. Wine Storytelling

Moscardo [51] states that central themes and stories impact on tourists and their behaviour. Winery visits by tourists provide wine producers with a communication platform for their brand's stories, while also showcasing their product portfolio [52]. Winemakers may tell many stories about the wine production: Their families, their heritage, and their winemaking approach. The wine tourist may also evaluate the stories when deciding which wine to buy [53]. Wine-related stories become

part of the wine experience and may be relived by repeating the story [54]. As storytelling allows consumers to integrate the story of a wine brand or property [55] and enhance their wine experience, this element should also be measured, as storytelling value adds to the wine tourism experience.

## 2.6. Wine Involvement

According to O'Neill and Charters [52] winery visits increase the direct involvement with the tourist. The relationship between consumers' travel and their involvement with wine proves their strong dependence [11,56]. Wine tourism and involvement with wine are described as a consumer experience with a high hedonic charge [11]. Brown, Havitz, and Getz [57] found that the particular interest in a product (wine) has the effect of creating the desire to travel to the place where the product is made. Wine consumers' product involvement is also equated with their own personal involvement with wine [58]. Yuan et al. [59] maintain that wine consumers' feelings of importance and relevance towards a product, as well as their genuine level of interest in wine, are determined through a high level of product involvement. Bruwer and Alant [23] offer the view that the wine tourist is drawn to be involved with the wine and region where the wine is produced. Engagement by individuals in wine tourism is related to a desire to become better acquainted with the wine product and to enjoy an indulgent experience [23]. Sthapit et al. [60] attest that involvement is one of seven experiential tourism factors, significantly influencing the memorability of the tourists' experience.

Wine and wine tourism provide and drive a set of authentic and genuine experiences for wine tourists, which are increasingly differentiated and personalised [61]. Thus, the wine tourism experience is an amalgam of components and features related to wine, with dimensions such as wine excitement, wine sensory appeal, winescape, and wine involvement, which play a crucial role in the wine tourists' experience.

## 3. Research Method

### 3.1. Scale Development Process

Scale validity refers to the degree to which a study accurately reflects or assesses the specific concept that the researcher is attempting to measure, while reliability refers to the degree to which a test is consistent and stable in measuring what it is intended to measure [62]. Consequently, to ensure the reliability and validity of the methods used to construct and validate the scale in this study, four aspects were taken into account: (1) Domain of construct, (2) item generation, (3) purifying the measurement, and (4) finalising the measurement [62,63], comprising the scale development process through the major methodological stages which focus on the scale development process.

### 3.2. Item Generation

Derived from several studies (Table 1), an initial pool of 20 items was constructed and generated. The initial items were then refined and edited for content validity by six experts in related academic or practical fields. With the intention of classifying the items into construct groups, a sorting procedure was used by the experts to refine items that were considered redundant or ambiguous. The items were not grouped or sequenced, and only one conceptual change resulted from the process where the experts found it difficult to distinguish between 'wine sensory appeal' and 'wine excitement', this being replaced by 'wine tasting excitement'. The process resulted in 18 modified measurement items, classified into four categories: Wine tasting excitement, winescape, wine storytelling, and wine involvement (Table 1).

The 18-item instrument was pretested with a convenience sample of 65 participants who had a wine experience at Porto and Madeira wine cellars, as wine tourism destinations, during July 2019. The goal of this pre-test was to identify possible weaknesses, ambiguities, missing and redundant questions, and poor reliability [62]. As Netemeyer et al. [64] argue, the construct validity can be supported by this process, as the exclusion of items that may be conceptually inconsistent is allowed.

To determine the scale dimensions, exploratory factor analysis (EFA) was performed, which is a preliminary technique in the scale development process and construct validation [65]. An inspection of the strength of the relationship between the items is necessary to assess whether a particular data set is suitable for factor analysis [66]. It was found that no items had factor loadings lower than 0.4 or cross-loaded on more than one factor. A Cronbach's alpha reliability score higher than 0.7 indicated that the variables exhibited moderate correlation with their factor groupings and were regarded as internally consistent and stable [66]. As a result, no items had factor loadings lower than 0.4 or cross-loaded on more than one factor, and therefore no item was eliminated [66]. A total of 18 items with four constructs remained: Wine tasting excitement, winescape, wine storytelling, and wine involvement. A confirmatory factor analysis (CFA) analysis was then performed to confirm the structure of the scale. Moreover, CFA also evaluates the relationships between observed measures or indicators and latent variables or factors in detail [65]. CFA was applied, allowing free correlations for the whole sample and for a randomly split subsample. Convergent and discriminant analysis were used to test the scale as well as model fit. The last steps were to test a second-order factor analysis and then the multigroup analysis was applied.

**Table 1.** Initial scale items of wine experience.

| Dimensions | Scale Items Adjusted to Wine Experience | Support References |
|---|---|---|
| Wine Tasting Excitement | 1. Tasting this wine in its original wine cellars makes me excited<br>2. Tasting this wine on holidays helps me to relax<br>3. Tasting this wine makes me feel exhilarated<br>4. Tasting this wine on holidays makes me stop worrying about routine | [41,67,68]. |
| Winescape | 5. This winery landscape has a rural appeal<br>6. These buildings have historic appeal<br>7. There is an old-world charm in these wine cellars<br>8. This architecture gives the winery character | [14,23,56,68,69]. |
| Wine Storytelling | 9. Stories told about the wine positively influenced the value I attribute to it<br>10. Stories told about the wine positively influenced the value I attribute to the wine tasting<br>11. Stories told about the wine positively influenced the value I attribute to this visit<br>12. Stories told about the wine enabled me to have an enjoyable time<br>13. Stories told about the wine enabled me to learn ancient facts about wine that I did not know | [53,54,70]. |
| Wine Involvement | 14. I like to purchase wine to match the occasion<br>15. For me, drinking this wine gives me pleasure<br>16. I enjoyed these wine activities which I really wanted to do<br>17. For me, these wine tastings are a particularly pleasurable experience<br>18. My interest in this wine makes me want to visit these wine cellars | [57,71]. |

*3.3. Purifying the Measurement*

The list of resulting measurement items was verified with 379 wine tourists who had visited Madeira and Porto wine cellars, and these items were measured using a seven-point Likert scale, varying from 1 (strongly disagree) to 7 (strongly agree). The final survey (multilingual: English, Spanish, French, and Portuguese) was administered by the researcher to a convenience sample of wine tourists visiting Porto and Madeira wine cellars between July and September 2019. The data analysis was carried out in two stages: An (1) EFA, followed by a (2) confirmatory factor analysis (CFA), using SPSS (version 26) and AMOS (version 26). An exploratory factor analysis (EFA) using the generalised least squares as extraction method with a varimax rotation and Kaiser normalisation was undertaken

on the data collected to determine the dimensions of the scale. The criteria used to extract factors was an eigenvalue > 1. The EFA was run separately for each factor.

The EFA identified four dimensions, explaining 58.94% of overall variance, labelled: (1) Wine tasting excitement, (2) winescape, (3) wine storytelling, and (4) wine involvement. Both Bartlett's test of sphericity (a statistical test for the presence of correlations among the variables) and the Kaiser–Meyer–Olkin (KMO) measure of sampling adequacy were measured to assess data factorability. A KMO value of 0.942 exceeds the acceptable minimum value, which is 0.6 [66]. Bartlett's test of sphericity was found to be significant ($p < 0.000$), within the recommended boundaries (Table 2). The findings presented Cronbach reliability scores ranging from 0.86 to 0.92. In addition, during the factor extraction process, no items were removed. Factor loadings were not revealed to be cross-loaded on different factors, and therefore no item was eliminated.

**Table 2.** Exploratory factor analysis results for the initial measurement scale (wine tourists $n = 647$).

| Dimensions and Items | Factor Loading | Mean | SD | Total Variance Explained (%) | Cronbach's Alpha |
|---|---|---|---|---|---|
| **Wine Tasting Excitement** | - | - | - | 16.662 | 0.887 |
| 1. Tasting this wine in its original wine cellars makes me excited | 0.697 | 6.23 | 1.074 | - | - |
| 2. Tasting this wine on holidays helps me to relax | 0.688 | 5.90 | 1.281 | - | - |
| 3. Tasting this wine makes me feel exhilarated | 0.725 | 5.91 | 1.256 | - | - |
| 4. Tasting this wine on holidays makes me stop worrying about routine | 0.658 | 5.88 | 1.414 | - | - |
| **Wine Storytelling** | - | - | - | 15.952 | 0.888 |
| 1. Stories that the wine tour guide/winemaker/wine producer told about the wine positively influenced the value I attribute to the wine | 0.819 | 6.30 | 1.014 | - | - |
| 2. Stories that the wine tour guide/winemaker/wine producer told about the wine positively influenced the value I attribute to the wine tasting | 0.770 | 6.15 | 0.977 | - | - |
| 3. Stories that the wine tour guide/winemaker/wine producer told about the wine positively influenced the value I attribute to this visit | 0.703 | 6.21 | 0.882 | - | - |
| 4. Stories that the wine tour guide/winemaker/wine producer told about the wine enabled me to have an enjoyable time | 0.689 | 6.22 | 0.916 | - | - |
| 5. Stories that the wine tour guide/winemaker/wine producer told about the wine enabled me to learn ancient facts about wine that I did not know | 0.691 | 6.30 | 1.029 | - | - |
| **Wine Involvement** | - | - | - | 14.442 | 0.876 |
| 1. I like to purchase wine to match the occasion | 0.626 | 6.16 | 1.071 | - | - |
| 2. For me, drinking this wine gives me pleasure | 0.677 | 6.33 | 0.886 | - | - |
| 3. I enjoyed these wine activities which I really wanted to do | 0.689 | 6.19 | 0.926 | - | - |
| 4. For me, these wine tastings are a particularly pleasurable experience | 0.699 | 6.34 | 0.857 | - | - |
| 5. My interest in this wine makes me want to visit these wine cellars | 0.534 | 6.27 | 1.012 | - | - |
| **Winescape** | - | - | - | 11.880 | 0.793 |
| 1. This winery landscape has a rural appeal | 0.570 | 6.20 | 1.017 | - | - |
| 2. These buildings have historic appeal | 0.642 | 6.40 | 0.846 | - | - |
| 3. There is an old-world charm in these wine cellars | 0.705 | 6.27 | 0.868 | - | - |
| 4. This architecture gives the winery character | 0.585 | 6.32 | 0.855 | - | - |

KMO: 0.942, Bartlett's test of sphericity: 7860.099, Sig.: 0.000

## 4. Results and Discussion

### 4.1. Sample Profile

The sample (Table 3) was balanced in terms of gender, with most visitors from the United Kingdom, France, Portugal, or Germany, and the majority being adults between 25 and 54. The sample had high education levels and a medium- to high-level job standard, and represented the main market in Portugal.

**Table 3.** Socio-demographic profile of the sample—whole data (*n* = 647).

| Gender | Age | Education Level | Country of Origin | Job |
|---|---|---|---|---|
| Male (49.7%) | 18–24 years old (7.1%) | Less than high school graduate (3.7%) | Portugal (8.3%) Spain (5.6%) | Businessperson/manager (16%) Freelancer/self-employed (17.9%) |
| | 25–34 years old (21.3%) | | | Middle/senior employed |
| | 35–44 years old (21%) | High school graduate (18.5%) | France (24.7%) Germany (7.7%) | management (17%) Civil servant (11.4%) |
| Female (50.3%) | 45–54 years old (27.8%) | Degree (43.8%) Master's degree | United Kingdom (25.9%) | Worker (17.4%) Pensioner/retired (4%) |
| | 55–64 years old (16%) | (27.2%) Doctorate (6.8%) | Other countries (27.8%) | Domestic/unemployed (1.5%) Student (6.5%) |
| | 65 or > years old (6.8%) | | | Other (8.3%) |

### 4.2. Finalising the Measurement

Further robust and consistent data collection was carried out to assess the reliability and validity of the measurement scale. Likewise, the data gathered from the sample of wine tourists recruited in Madeira and Porto wine cellars (*n* = 647) was used to accomplish the CFA, because the development sample must be sufficiently large [62,64]. In total, 323 responses were collected in Madeira wine cellars and 324 responses were collected in Porto wine cellars between late July and September 2019 (the high season). Therefore, a total of 647 self-administrated questionnaires were considered valid and usable for data analysis.

The confirmatory factor analysis (CFA) was conducted using the generalised least squares method [72,73]. to assess the validity and reliability of the constructs. As result, 18 indicators were retained for inclusion in the final scale (Table 4). The adjustment results improved significantly, yielding the values in Table 4 and the adjustment values expressed. As concerns validity and reliability, for the average variance extracted (AVE), the value obtained also exceeds the reference cut-off value (≥0.50) according to the literature [66,70] (Table 5).

The overall goodness-of-fit index (Table 5) displayed a suitable level of fit: $\chi^2$ = 406.302; df = 129; $p$ = 0.000; $\chi^2$/df = 3.15; GFI = 0.93; AGFI = 0.907; RMSEA = 0.058, with the result in keeping with what is suggested in the literature [66], confirming the scale's goodness of fit. These results suggest that the proposed model fits well with the empirical data. This study represents one of the first major efforts to propose wine experience factors at wine tourism destinations and, following the accepted scale development procedure [62,64]. developed a measurement scale for wine experience. The final analysis to validate the scale comprises wine storytelling (5 items), wine involvement (5 items), winescape (4 items), and wine tasting excitement (4 items).

**Table 4.** Confirmatory factor analysis results for final measurement scale (wine tourists *n* = 647).

| Constructs and Indicators | | | St. Regression | S.E. | C.R. | *p* |
|---|---|---|---|---|---|---|
| Stories that the wine tour guide/winemaker/wine producer told about the wine enabled me to learn ancient facts about wine that I did not know | <— | Wine Storytelling | 0.798 | - | - | - |
| Stories that the wine tour guide/winemaker/wine producer told about the wine enabled me to have an enjoyable time | <— | Wine Storytelling | 0.848 | 0.042 | 23.778 | *** |
| Stories that the wine tour guide/winemaker/wine producer told about wine positively influenced the value I attribute to this visit | <— | Wine Storytelling | 0.826 | 0.044 | 22.402 | *** |
| Stories that the wine tour guide/winemaker/wine producer told about the wine positively influenced the value I attribute to the wine tasting | <— | Wine Storytelling | 0.88 | 0.05 | 22.376 | *** |
| Stories that the wine tour guide/winemaker/wine producer told about the wine positively influenced the value I attribute to the wine | <— | Wine Storytelling | 0.891 | 0.045 | 25.063 | *** |
| Tasting this wine on holidays makes me stop worrying about routine | <— | Wine Tasting Excitement | 0.84 | - | - | - |
| Tasting this wine makes me feel exhilarated | <— | Wine Tasting Excitement | 0.808 | 0.046 | 23.441 | *** |
| Tasting this wine on holidays helps me to relax | <— | Wine Tasting Excitement | 0.874 | 0.043 | 25.592 | *** |
| Tasting this wine in its original wine cellars makes me excited | <— | Wine Tasting Excitement | 0.79 | 0.035 | 22.06 | *** |
| My interest in this wine makes me want to visit these wine cellars | <— | Wine Involvement | 0.773 | - | - | - |
| For me, these wine tastings are a particularly pleasurable experience | <— | Wine Involvement | 0.846 | 0.044 | 21.931 | *** |
| I enjoyed these wine activities which I really wanted to do | <— | Wine Involvement | 0.833 | 0.05 | 21.069 | *** |
| For me, drinking this wine gives me pleasure | <— | Wine Involvement | 0.837 | 0.048 | 20.021 | *** |
| I like to purchase wine to match the occasion | <— | Wine Involvement | 0.841 | 0.056 | 21.02 | *** |
| This winery landscape has a rural appeal | <— | Winescape | 0.75 | - | - | - |
| These buildings have historic appeal | <— | Winescape | 0.78 | 0.052 | 17.594 | *** |
| There is an old-world charm in these wine cellars | <— | Winescape | 0.805 | 0.056 | 17.29 | *** |
| This architecture gives the winery character | <— | Winescape | 0.82 | 0.055 | 17.62 | *** |

Notes: *** *p*-value < 0.01.

**Table 5.** Goodness-of-fit indexes for the measurement.

| Dimensions | CR | AVE | MSV | ASV | Wine Involvement | Wine Storytelling | Wine Tasting Excitement | Winescape |
|---|---|---|---|---|---|---|---|---|
| Wine Involvement | 0.915 | 0.683 | 0.594 | 0.570 | 0.826 | - | - | - |
| Wine Storytelling | 0.928 | 0.721 | 0.527 | 0.479 | 0.726 | 0.849 | - | - |
| Wine Tasting Excitement | 0.897 | 0.687 | 0.594 | 0.537 | 0.771 | 0.690 | 0.829 | - |
| Winescape | 0.868 | 0.623 | 0.590 | 0.521 | 0.768 | 0.659 | 0.735 | 0.789 |
| GOF Indexes | - | X2 | Df | *p*-value | X2/df | GFI | AGFI | RMSEA |
| Whole sample (*n* = 647) | - | 406.302 | 129 | *** | 3.15 | 0.93 | 0.907 | 0.058 |

Notes: *** *p*-value < 0.01.

The structural equation model and values of standardised structural coefficients are shown in Figure 1. It was proven by the statistical analysis that all dimensions contribute to the definition of the wine experience construct. The evaluation of the significance of a regression coefficient was performed by an analysis of the *t*-test [74]. The existence of a significant regression coefficient (the value of t exceeds 1.645) assumed that the relationship between the two latent variables was demonstrated empirically [66]. In addition, the case of a positive or satisfactory evaluation of adjustment measures confirmed the predictive validity of the model [74]. In this study, it was assumed that in unilateral cases (direct and positive influence), significant relations would present a t-value of greater than 1.645. Overall, the data supported that wine experience was explained by the four latent factors: Wine storytelling, wine involvement, winescape, and wine tasting excitement.

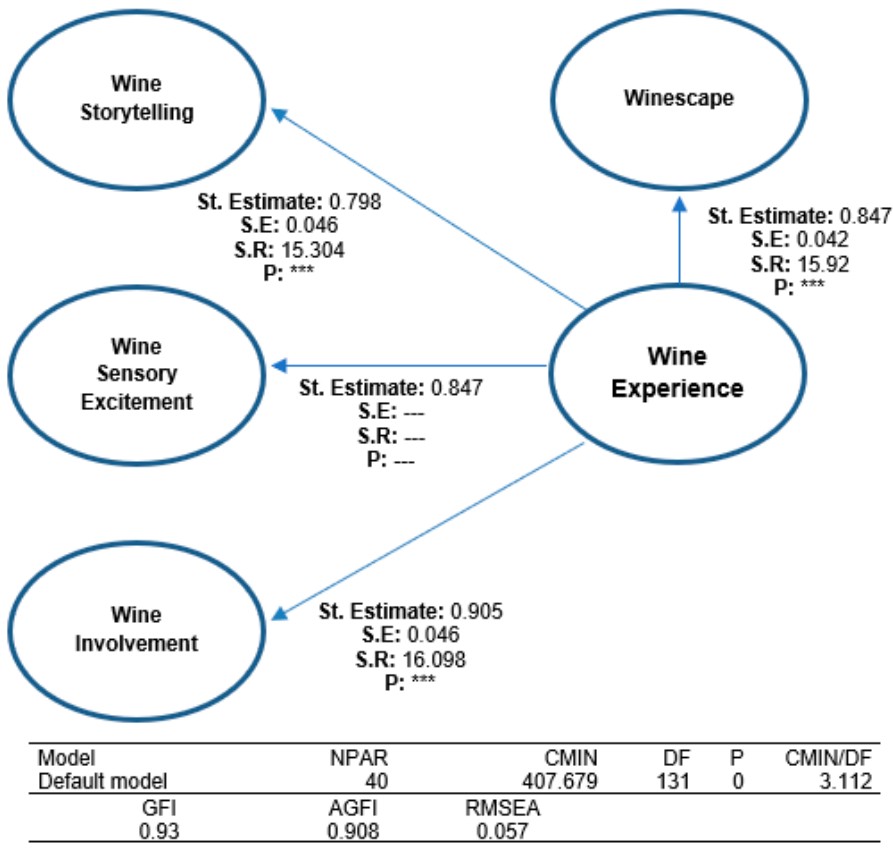

| Model | NPAR | CMIN | DF | P | CMIN/DF |
|---|---|---|---|---|---|
| Default model | 40 | 407.679 | 131 | 0 | 3.112 |
| GFI | AGFI | RMSEA | | | |
| 0.93 | 0.908 | 0.057 | | | |

**Figure 1.** Structural equation model of final measurement scale. *** *p*-value < 0.01.

Following the SEM analysis, variable correlations were tested for invariance among two different groups of wine tourists. A multigroup analysis (Table 6) highlighted how the Porto and Madeira wine cellars differ from each other from the wine tourists' perspective within these two wine tourism destinations, based on the proposed scale. Overall, the findings supported all the hypothesised relationships in both tourism destinations, which reinforces the consistency of the wine experience scale. The two main differences were wine storytelling and winescape. Wine storytelling by the wine tourists was more evident in Madeira (0.718, $p < 0.05$) than in Porto (0.574, $p < 0.05$). It is expected that this discrepancy was related to greater personalisation of the guided wine tours in Madeira wine cellars as compared to Porto wine cellars. The winescape was more evident in Porto (0.696, $p < 0.05$) than in Madeira (0.655, $p < 0.05$), probably due to the cellar landscape, scenery, ancient architecture, and panoramic views around the cellars.

**Table 6.** Multi group analysis.

| Dimensions and Constructs | - | - | Porto Wine Cellars | | Madeira Wine Cellars | | - |
|---|---|---|---|---|---|---|---|
| - | - | - | Estimate | *p* | Estimate | *p* | z-Score |
| Wine Storytelling | <— | Wine Experience | 0.574 | *** | 0.718 | *** | 1.72 * |
| Wine Involvement | <— | Wine Experience | 0.720 | *** | 0.696 | *** | −0.273 |
| Winescape | <— | Wine Experience | 0.696 | *** | 0.655 | *** | −0.501 |
| Wine Tasting Excitement | <— | Wine Experience | 1.000 | - | 1.000 | - | - |
| Wine Storytelling 5 | <— | Wine Storytelling | 1.000 | - | 1.000 | - | - |
| Wine Storytelling 4 | <— | Wine Storytelling | 1.137 | *** | 0.949 | 0.000 | −1.942 * |
| Wine Storytelling 3 | <— | Wine Storytelling | 1.092 | *** | 0.977 | 0.000 | −1.149 |
| Wine Storytelling 2 | <— | Wine Storytelling | 1.314 | *** | 1.050 | 0.000 | −2.273 ** |
| Wine Storytelling 1 | <— | Wine Storytelling | 1.327 | *** | 1.011 | 0.000 | −3.027 *** |
| Wine Tasting Excitement 3 | <— | Wine Tasting Excitement | 1.000 | | 1.000 | | |
| Wine Tasting Excitement 2 | <— | Wine Tasting Excitement | 1.124 | *** | 1.011 | 0.000 | −1.349 |
| Wine Tasting Excitement 1 | <— | Wine Tasting Excitement | 0.807 | *** | 0.718 | 0.000 | −1.261 |
| Wine Involvement 5 | <— | Wine Involvement | 1.000 | - | 1.000 | - | - |
| Wine Involvement 4 | <— | Wine Involvement | 0.991 | *** | 0.953 | 0.000 | −0.423 |
| Wine Involvement 2 | <— | Wine Involvement | 1.031 | *** | 1.091 | 0.000 | 0.589 |
| Wine Involvement 2 | <— | Wine Involvement | 0.885 | *** | 1.045 | 0.000 | 1.652 * |
| Wine Involvement 1 | <— | Wine Involvement | 1.240 | *** | 1.086 | 0.000 | −1.326 |
| Winescape 1 | <— | Winescape | 1.000 | - | 1.000 | - | - |
| Winescape 2 | <— | Winescape | 1.058 | *** | 0.668 | 0.000 | −3.895 *** |
| Winescape 3 | <— | Winescape | 1.074 | *** | 0.693 | 0.000 | −3.587 *** |
| Winescape 4 | <— | Winescape | 0.944 | *** | 0.850 | 0.000 | −0.906 |
| Wine Tasting Excitement 4 | <— | Wine Tasting Excitement | 1.076 | *** | 1.040 | 0.000 | −0.400 |

Notes: *** *p*-value < 0.01; ** *p*-value < 0.05; * *p*-value < 0.10.

Advancing these results, meaningful conclusions were drawn and explained, and confirm that the dimensions focus on experiential wine tourism in a holistic way, directly demonstrated by the nature of their corresponding items. Thereby, the wine experience is shaped by four dimensions (wine storytelling, wine tasting excitement, wine involvement, and winescape), directly correlated between them in a composite way, justifying their inclusion on the same scale. Moreover, the results identified dimensions with stronger relevance and impact; foremost was wine storytelling, followed by wine involvement and wine tasting excitement (both very close), and finally winescape. These statements underline the premise value of holistic and hedonic wine experience and yield valuable insights through the increased participation of the wine tourists in the visits. Asero and Patti [75] regarded wine as a decoy that attracted visitors, considering it the soul of the wine tourism, and that it is an experience derived from the hedonic nature of wine tasting [76]. The wine experience dimensions (wine storytelling, wine tasting excitement, wine involvement, and winescape) fulfil a congruent logic that is undoubtedly justified by the relationship between them as the results suggest. The research results highlight the relevance of these dimensions to provide and guarantee an immersive experience to offer a "best holistic wine experience" to wine tourists and potential visitors. It is noteworthy that the wine tourists appreciate a holistic tourism experience due to interactions with other wine visitors and winery staff [76]. Moreover, these findings align with several studies [3,7,19,22,39,77,78].

## 5. Conclusions

If tourism is to succeed and expand in the future, new paradigms have to be brought into the field [79]. Wine tourism as a form of tourism may make a great contribution to the tourism industry and to the development of new experiential paradigms. Such experiences are often offered in small-sized, rural establishments that are linked to nature and offer social distancing. This study has established a reliable and valid 18-item scale composed of four dimensions to measure the wine experience within a wine tourism context. This was applied in two different environments and with both national and international wine tourists. The research clearly highlights the major finding that the wine experience construct is formed by the four dimensions proposed (wine storytelling, wine tasting excitement, wine involvement, and winescape) that simultaneously and accurately depict the wine-related tourism

experience as being justified by the significant relationships between dimensions. Wine storytelling appears as a most significant dimension due to the fact that visits to the wine cellars begin and end with the wine tour guide/winemaker/wine producer, where there are authentic stories related to wine and wine tourism, which are much appreciated by wine tourists. The tasting of the wine also creates delight through a wine sensorial excitatory stimulus. It is also common for wine tourists to be involved during the visits where wine tourists appreciate the wine scenery in the cellar winescape.

This is the first study demonstrating the combined used of wine experience dimensions in constructing a useful measurement tool. This measurement approach extends this scope because a scale had never been developed to measure the wine experience of wine tourists. Hence, there was the emerging need for development and validation this new scale—the wine experience scale. The study also reinforces the growing literature on wine experience by establishing representative constructs which address research gaps in terms of the lack of a validated scale to evaluate the wine experience. Therefore, the measurement tool proposed in this study provides a procedure for further examination in future wine tourism research. The wine experience dimensions within the wine tourist experience are an important topic in wine tourism research, and thus these dimensions are considered key wine experience drivers, derived from empirical evidence and a holistic approach, understood as essential to more successful and memorable wine experiences for all kinds of wine tourists. It is recognised that wine tourists expect the "best wine experience". The consistent relationship between the four underlying dimensions was demonstrated, and it was proven that, as a whole, they form the wine experience construct. The main management implications imply that managers should understand how a wine tour experience can be improved across a range of wine dimensions in a highly immersive wine experience, as is the case and example of wine and cultural heritage [80]. Wine tour guides should take full advantage of their close contact with wine tourists during the visits and should be monitored and more customized, first to reflect on better performance in wine guided tours, and thus achieving a better wine engagement in the future. Hence, exclusive and memorable wine experiences can be promoted as follows: Wine tours, wine tastings, wine events, and wine courses, among others, taking full advantage of the kind of wine tourist profile (e.g., wine lovers, wine interested, and wine curious), according to other studies similar to this one [81].

The results further underline the importance of wine as the main core product in wine tourism experiences. In addition, wine tourists in Madeira and Porto wine cellars retain quite strong, distinctive impressions of each wine cellar-related travel. Notably, there is a growing potential for managers, stakeholders, players, opinion makers/leaders, and marketers to extract benefits from this managerial point of view [82]. In summary, wine potentiates a multi-experience for wine tourists, so managers can get better results by designing wine and wine tourism products and communication strategies around the main themes linked to each dimension of the wine experience explained in this research, all of which appear to contribute to more complete wine experiences.

Regarding the research limitations and suggestions for future research, the period of data collection (during the summer) was short, although it is the time when there is the largest number of wine tourists, which leads to the suggestion for a cross sectional evaluation of wine experience of the wine tourists, for instance every season, and a comparison with other cellars of the new- and old-world wine tourism destinations. In this way, high coverage of the population can be achieved in order to establish the generalisability and consistency of this newly developed and validated scale.

**Author Contributions:** Conceptualization, V.S., P.R., and N.A.; methodology, V.S. and P.R.; validation, V.S. and P.R.; formal Analysis, V.S. and P.R.; investigation, V.S and P.R.; resources, V.S. and N.A.; data curation, V.S. and P.R.; writing—original draft, P.R., V.S., N.A., and E.S.-P. Writing—review & editing, V.S. and P.R. All authors have read and agreed to the published version of the manuscript.

**Funding:** This research was funded by national funds through FCT—Foundation for Science and Technology, IP, within the scope of the reference project UIDB/04470/2020.

**Acknowledgments:** CiTUR—Research, Development and Innovation Center, Polytechnic of Leiria and ISLA Santarém—Higher Institute of Management and Administration of Santarém.

**Conflicts of Interest:** The authors declare no conflict of interest.

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
