# Peer review of "Developing a Wine Experience Scale: A New Strategy to Measure Holistic Behaviour of Wine Tourists"

_sustainability, doi:10.3390/su12198055_

Round 1

Reviewer 1 Report

The paper aims to measure wine tourism experiences, in the cases of two fortified wine tourism regions and destinations in Portugal (Porto and Madeira), by using a panel of 647 wine tourists.
The major strength of this study is that the impact of antecedents of wine experience, by taking into consideration four dimensions, is an important issue in (Portuguese) wine literature. On the other hand, it presents some points that should be improved, which have been thoroughly listed below.

Author Response

All required improvements/minor revisions were considered.

Reviewer 2 Report

I think it is an interesting and innovative article that applies a rigorous methodology and shows consistent results.

Author Response

There are no requested revisions.

Reviewer 3 Report

I commend the authors on an exceptional paper. Theory is progressed and there are relevant industry applications. The literature review is on point and analysis is solid. 

Author Response

There are no requested revisions.